# Phytochemical Profile, Safety and Efficacy of a Herbal Mixture Used for Contraception by Traditional Health Practitioners in Ngaka Modiri Molema District Municipality, South Africa

**DOI:** 10.3390/plants11020193

**Published:** 2022-01-12

**Authors:** Molelekwa Arthur Moroole, Simeon Albert Materechera, Wilfred Otang-Mbeng, Rose Hayeshi, Cor Bester, Adeyemi Oladapo Aremu

**Affiliations:** 1Indigenous Knowledge Systems Centre, Faculty of Natural and Agricultural Sciences, North-West University, Private Bag X2046, Mmabatho 2790, North West Province, South Africa; albert.materechera@nwu.ac.za; 2School of Biology and Environmental Sciences, Faculty of Natural Sciences and Agriculture, University of Mpumalanga, Mbombela Campus, Private Bag X11283, Nelspruit 1200, Mpumalanga Province, South Africa; wilfred.mbeng@ump.ac.za; 3Department of Science and Innovation/North-West University Preclinical Drug Development Platform (PCDDP), Faculty of Health Sciences, North-West University, Private Bag X6001, Potchefstroom 2520, North West Province, South Africa; rose.hayeshi@nwu.ac.za (R.H.); cor.bester@nwu.ac.za (C.B.)

**Keywords:** acute toxicity, contraceptive, cytotoxicity, medicinal plants, phenolics, toxicity

## Abstract

The use of medicinal plants for contraception remains a common practice among South African ethnic groups. The present study assessed the phytochemical profile, cytotoxicity, acute oral toxicity and efficacy of a herbal mixture used for contraception by the Batswana of South Africa. An aqueous extract was prepared from equal quantities (in terms of weight) of *Bulbine frutescens* (roots), *Helichrysum caespititium* (leaves) and *Teucrium trifidum* (leaves) based on a recipe used by traditional health practitioners. The phytochemical profiles of the freeze-dried herbal mixture were analyzed using gas chromatography–mass spectrometry (GC-MS). In addition, cytotoxicity was determined using an MTT assay on Vero cells and in vivo contraceptive efficacy was evaluated using seven Sprague Dawley rats per control and treatment groups. The control group received distilled water while test groups received 5, 50 and 300 mg/kg of the herbal mixture, which was administered orally once a day for three consecutive days. Subsequently, female rats were paired 1:1 with males for 3 days. Their weights were measured weekly and incidence of pregnancy was recorded. The GC-MS chromatogram revealed the presence of 12 identified and 9 unidentified compounds. In terms of safety, the herbal mixture had an IC_50_ value of 755.2 μg/mL and 2000 mg/kg, which was the highest tested dose that caused no mortality or morbidity in the rats. A contraceptive efficacy of 14.5% was exerted with 50 mg/kg herbal mixture extract while other doses had no effects given that all the rats were pregnant. Based on a chi-square test (*p* < 0.05), there was no correlation between the tested herbal mixture doses and contraception, nor on the weight of the rats. Overall, the herbal mixture extract was found to be safe but had limited contraceptive efficacy at the tested doses. In future studies, exploring increased dose range, solvent extract types and hormonal analysis will be pertinent.

## 1. Introduction

Globally, the potential of various parts and extracts of medicinal plants as contraceptive agents is gaining increasing interest [1,2,3,4,5,6]. On the basis of existing anecdotal evidence, researchers have explored natural resources, including plants, as alternative to conventional conceptive [3,7,8]. As an indication of the unlocked potential in medicinal plants, recent studies have efficiently established *Azadirachta indica* as an effective contraceptive agent [9]. In African societies, the use of plants for purposes of regulating fertility has been practiced for centuries, thereby highlighting their potential as contraceptives [7,10]. In Tanzania, several herbalists confirmed the existence of some herbs that might act as postnatal contraceptives when taken for several weeks after the delivery of a baby [11]. Likewise, similar evidence on the dependence on plants and associated herbal products has been reported in several African countries such as Gambia [12], Nigeria [13], South Africa [7] and Zimbabwe [14]. Examples of common plants used in these aforementioned countries include *Bulbine latifolia*, *Cannabis sativa*, *Pouzolzia mixta*, *Securidaca longipedunculata*, *Withania somnifera* and *Ziziphus mucronata*.

In South Africa, medicinal plants used for contraception showed that herbal contraceptives are usually administered orally as single species or as herbal mixtures [7,15,16]. Based on indigenous knowledge among different ethnic groups, the use of herbal mixtures or concoctions is often geared at enhancing the effectiveness and safety of the herbal product [17,18]. The importance of establishing the phytochemical profiles, safety and biological efficacy of herbal mixtures cannot be overemphasized [17,19,20,21,22].

Recently, we documented the use of medicinal plants as an important approach for preventing pregnancy among the Batswana in Ngaka Modiri Molema District Municipality, North West province, South Africa [23]. Particularly, a herbal mixture consisting of the roots (*Bulbine frutescens* (L.)) and leaves (*Helichrysum caespititium* (DC.) Harv. and *Teucrium trifidum* Retz.) is considered as a popular remedy for preventing pregnancy. However, the chemical composition, safety and efficacy of the herbal mixture remain unknown. Thus, this study assessed the phytochemical profile, cytotoxicity, acute oral toxicity and efficacy of the herbal mixture used for contraception by Batswana traditional health practitioners in Ngaka Modiri Molema District Municipality of the North West province, South Africa.

## 2. Materials and Methods

### 2.1. Collection and Identification of Voucher Specimens

A traditional health practitioner with knowledge and experience of medicinal plants used for contraception in the study area assisted with the collection of the three plant species between August and September 2019. Voucher specimens were sent to the South African National Biodiversity Institute (SANBI) in Pretoria for identification. In addition, voucher specimens were deposited at the S.P. Phalatse Herbarium of the North-West University (Mafikeng campus), North West Province, South Africa. The names and botanical families of species were validated taxonomically using the World Flora Online http://www.worldfloraonline.org (accessed on 29 December 2021).

### 2.2. Preparation of the Herbal Mixture

Individually, the three plants were washed under running tap water and shade dried for approximately 3–4 weeks. The samples were pulverized to powder using a Bennett Read 1200 W power mechanical blender. A kitchen sieve with 1 mm pores was used to remove solids. We combined equal portion (approximately 80 g each) of ground plant materials from *Bulbine frutescens* (roots), *Helichrysum caespititium* (leaves) and *Teucrium trifidum* (leaves). This was based on a recipe used by traditional health practitioners in Ngaka Modiri Molema District Municipality, North West province of South Africa (Table 1). A sample of the herbal mixture (100 g) was placed in a 2000 mL glass beaker with 2 l of water prior to boiling. The beaker was covered with a foil to prevent contamination, and allowed to cool and settle overnight. Thereafter, the mixture was centrifuged at low speed using a Sichuan Shuke lab model TD-600 Benchtop. The extract was filtered and lyophilized using a ViTis AdVantage 2.0 BenchTop Freeze Dryer/Lyophilizer operated at temperatures ranging from −55 °C to −85 °C. After 96 h, lyophilized plant material was stored in a dry 100 mL glass beaker tightly covered with foil to prevent contamination.

### 2.3. Phytochemical Profile of the Herbal Mixture

Phytochemical profile of the herbal mixture was determined using Gas chromatography–mass spectrometry (GC-MS), model GC-7890A/MS-5975C Agilent Technologies, Santa Clara, United States of America (USA). One milligram (1 mg) of the freeze-dried herbal mixture was weighed and dissolved in 1 mL of methanol. The GC capillary column (HP-5 MS, 30 m × 0.25 mm i.d., 0.25 μm) was used. High purity helium (99.996%) was used as carrier gas at a flow rate of 1.25 mL/min and a linear velocity of 37 cm/s. An aliquot of 1 µL of the sample was injected into the column with an inlet temperature of 250 °C and split-fewer modes at time of 30 s. An initial oven temperature of 35 °C was set and programmed to increase up to 285 °C at the rate of 10 °C per min with a holding time of 3 min at each increment. The ion source temperature maintained at 230 °C was employed while electron impact (EI) mass spectra were obtained at the acceleration energy of 70 eV. The mass acquisition range was 40–550 DA with a data acquisition rate of 10 spectra/S. The relative quantity of the chemical compounds present in the extract was expressed as a percentage based on peak area produced in the chromatogram. The compounds were identified by direct comparison of the mass spectrum of the analyte at a particular retention time to that of reference standards found in the National Institute of Standards and Technology (NIST) library. The area percentage of each component was calculated by comparing its average peak area to the total areas obtained.

### 2.4. In Vitro Cytotoxicity Test Based on 3-(4,5-Dimethylthiazol-2-yl)-2,5-Diphenyl Tetrazolium Bromide (MTT) Assay

The cytotoxicity of the herbal mixture was evaluated using the Vero cell line (African green monkey) as described by Mosmann [24]. Cells at passages 11–12 were seeded at 3000 cells per well of a 96-well plate and incubated for 24 h in 5% CO_2_ at 37 °C, to adhere to the surface. The cells were treated every 24 h, at serial dilutions of concentration ranging between 5 μg/mL and 2000 μg/mL for a duration of 72 h before the cytotoxicity assay was initiated. The herbal mixture extract was dissolved in complete culture media at 2000 μg/mL, filtered through a 0.22 μm syringe filter, from which working solutions were prepared in complete culture medium prior to treatments. At the end of the incubation period, treated cells were rinsed twice with 100 μL phosphate-buffered saline (PBS), and 200 μL MTT (3-(4,5-dimethylthiazol-2-yl)-2,5-diphenyl tetrazolium bromide) solutions at a final concentration of 0.5 mg/mL was added to each well. Treated cultures were incubated for 4 h, after which excessive MTT solution was removed and discarded. Subsequently, 200 μL of Dimethylsulfoxide (DMSO) was added to dissolve the water insoluble formazan and the plate was incubated for 1 h. Before measurement of absorbance, the plate was placed on the shaker for 15 min and absorbance was measured at 560 and 630 nm.

The herbal mixture was tested on three technical repeats and two biological replicates. We used 0.2% Triton X-100 as the positive control. To measure the effect of TX-100, at the end of the experiment, untreated cells were rinsed following the same method as other experimental groups and exposed to TX-100 for 15 min prior to an addition of MTT solution. All MTT experiments were performed six-fold (two biological repeats of three technical repeats each). Cell survival rate expressed relative to untreated cell control was calculated as shown below:Cell survival rate (%) = (∆absorbance − ∆blank)/(∆control − ∆blank) × 100
where

Δabsorbance = absorbance at 560 nm − absorbance at 630 nm;

Δblank = DMSO blank at 560 nm – DMSO blank 630 nm;

Δcontrol = untreated cells at 560 nm − untreated cells at 630 nm.

### 2.5. Acute Oral Toxicity of Herbal Mixture

Acute oral toxicity refers to those adverse effects occurring following oral administration of a single dose of a substance, or multiple doses given within 24 h. The acute toxicity of the extract was determined as per the OECD guideline 423 [25]. The preferred species and strain was *Rattus norvegicus* (rat) and Sprague Dawley in particular. Healthy, non-pregnant rats (8–10 weeks) were randomly marked for identification and kept in cages 5 days prior to dosing to acclimatize to the laboratory conditions. The temperature in the experimental animal room was 22 °C (+3 °C) with relative humidity not exceeding 70% and lighting was artificial, the sequence being 12 h light and 12 h dark. For feeding, conventional laboratory chow was used with an unlimited supply of drinking water.

The 2000 mg/kg starting dose was selected from the required acute toxicity starting doses (5, 50, 300 or 2000 mg/kg) based on cytotoxicity results. Dose, used interchangeably with concentration in this study, refers to the amount of test substance administered and is expressed as weight of test substance per unit weight of test animal (e.g., mg/kg) [25]. The maximum volume of aqueous solution administered was 0.2 mL/100 g body weight and doses were prepared shortly prior to administration. Rats were fasted prior to dosing, with food but not water withheld overnight. Following the period of fasting, the animals were weighed and the herbal mixture administered. The herbal mixture was administered in a single dose orally using oral gavage. A 10% loss in body weight was considered critical and a threshold. The weight of each rat was measured 30 min, 2 h and 3 h after oral administration of the herbal mixture. Animals were thereafter observed for clinical signs of acute oral toxicity and their weight measured daily for 14 days. All observations and measurements were systematically recorded with individual records being maintained for each animal. At the end of the test, all animals were weighed and euthanized.

### 2.6. In Vivo Contraceptive Efficacy of Herbal Mixture

Healthy female Sprague Dawley rats of 8–10 weeks were kept in cages five days prior to dosing. The temperature, relative humidity, lighting, housing and care, feeding and supply of water were as described above. The efficacy study consisted of control and three test doses (5, 50 and 300 mg/kg) of herbal mixture extract with 7 rats per group as established by power analysis to be statistically viable. The three doses were selected because they were less than the lethal dose (LD_50_) of 2000 mg/kg determined from the acute oral toxicity study.

The rats in the control group had an average mass of 194.83 g while 5, 50 and 300 mg/kg treatments had an average weight of 197.93 g, 196.51 g and 191.75 g, respectively. The test herbal mixture extract was administered orally for three consecutive days. The female rats were caged with males at a ratio of 1:1 for three days at the end of administration of treatments. The weight of each female rat was measured before herbal administration and once after every seven days for 3 weeks. The animals were observed daily for 21 days and were all euthanized after the study. A formula for contraceptive efficacy was adapted from the study by Sewani-Rusike [16], as highlighted below:Contraceptive efficacy (CE) = number of non-pregnant rat/total number of rats × 100

### 2.7. Ethical Considerations

A plant collection permit for scientific curating and research was obtained from the North West Provincial Department of Rural Environment and Agricultural Development (READ). The animal (in vivo) experiment was approved by the North-West University Animal Care, Health and Safety Research Ethics Committee (NWU-AnimCareREC) withethics number NWU-00569-19-A5.

### 2.8. Data Analysis

SPSS statistical analysis software (IBM Analytics, Version 25), in conjunction with the Probit Analysis Method, was used to calculate IC_50_ (50% viability) values and 95% confidence limit ranges from the MTT analyses [26]. The Global Harmonized Classification System (GHS) was used to interpret the lethal dose (LD_50_) of the extract, and to categorize the acute toxicity of herbal mixture extract [25]. The chi-square test for independence was used to analyze correlations between contraception and concentrations (*p* < 0.05). The effect size (denoted by d) was used to analyze the effect of concentration on weight, where

d = 0.2 represented small effect and no practically significant difference;d = 0.5 represented medium effect and practically visible difference;d = 0.8 represented large effect and practically significant difference.

## 3. Results

### 3.1. Phytochemical Profile of the Herbal Mixture

The GC-MS chromatogram revealed the presence of 12 identified and 9 unidentified phytocompounds (Table 2). These included acetic acid, pyrrolidin-1-acetic acid, 2,3-butanedione, 2-propanone, 2,5-dimethyl-4-hydroxy-3(2H)-furarone and glycerin. Phenol, 4-ethyl and 2-methoxy-4-vinylphenol were detected with retention times of 1163.4 s and 1219.1 s, and peak area % of 0.2300 and 2.6723, respectively. The first compound to be detected was 2,3-butanedione with a retention time of 130.9 s. The least abundant compound was Benzeneethanol, 4-hydroxy- with a peak area of 0.0030054%. The most abundant compound was 2,3 butanediol with a peak area of approximately 44.7%.

### 3.2. Cytotoxicity of the Herbal Mixture

The tested herbal mixture extract had good solubility in complete media. Specifically, in the Vero cell line, the herbal mixture had none to very limited cytotoxicity up to concentrations of 300 μg/mL (Appendix A). From 700 μg/mL, toxicity appeared to increase in a concentration-dependent manner, with a loss of between 40% and 90% viability relative to the untreated control. The estimated IC_50_ value calculated for the tested herbal mixture extract in the Vero cell line was 755.2 μg/mL.

### 3.3. Acute Oral Toxicity of the Herbal Mixture

Animals were observed daily for 14 days post-dosing and the weight of each animal was measured. The first 3–4 h after administration of the extract was considered most important for acute toxicity studies [25]. After 2 h of herbal administration, the percentage in loss in body weight of animals was, on average, 2.07%, while the percentage loss in body weight after 3 h was 3.17 %, on average. However, only one rat showed vocalization when handled within the first 2 h after herbal administration. After 3 h of herbal administration, the weight of the rats decreased further by an average of 3% and all rats showed vocalization when handled. The average weight decreased from 193.61 g to 183.94 g within the first two days after extract administration. The percentage loss in body weight over the first two days after herbal extract administration was less than the 10% threshold. There was a general increase in average body weight over the next 12 days of the 14-day period (Figure 1).

No mortality or morbidity was observed after 14 days, an indication that the 2000 mg/kg dose was safe. Consequently, the lethal dose (LD_50_) value of the extract was interpreted to be between 2000 mg/kg to 5000 mg/kg, i.e., 2000 mg/kg < LD_50_ < 5000 mg/kg. Therefore, the herbal mixture extract was classified as a category 5 substance following the Globally Harmonized Classification System for Chemical Substances and Mixtures (GHS).

### 3.4. Contraceptive Efficacy of the Herbal Mixture

The weight of the rats in the control (0 mg/kg) increased by 56% on average, while the test groups, namely 5, 50 and 300 mg/kg, increased by 57%, 58% and 56%, respectively (Figure 2). Therefore, the highest increase in weight occurred in animals treated with 50 mg/kg herbal extract. Apart from 50 mg/kg treatment with 14.5% contraceptive efficacy, all the rats (in control, 5 and 300 mg/kg) fell pregnant, an indication of the absence of contraceptive effect. The chi-square test of independence did not show any correlation (*p* < 0.05) between concentration and contraception. Concentration had a minimal effect on weight with no practically significant difference.

## 4. Discussion

The most abundant compound was 2,3 butane-diol, but no information from the literature suggested that it could exert a contraceptive effect. Phenolic compounds are important bioactive compounds found in many medicinal plants [21,27,28]. In addition, phenolic profiling and evaluation of contraceptive effects on animals using contraceptive plants revealed the presence of phenols [6,8]. Furthermore, aryl 4-guanidinobenzoates that possess phenols have shown themselves to be potential vaginal contraceptives [29], and gossypol, a phenolic compound, has a reputed contraceptive effect [30]. Therefore, these two aforementioned phenolic compounds detected in the herbal mixture could be bioactive compounds responsible for contraceptive activity in the herbal mixture in traditional medicine, especially among the Batswanas of South Africa.

An investigation of *Helichrysum caespititium*, one of the species that formed the herbal mixture in our study, found the IC_50_ value of its extracts to be 428.77 μg/mL (hexane), 82.86 μg/mL (dichloromethane), 357.39 μg/mL (methanol) and 394.36 μg/mL (water) [31]. Some of the South African medicinal plants reported to be used for contraception have been investigated for cytotoxicity. For example, the IC_50_ of *Securidaca longipedunculata* was found to be 20.535 μg/mL [32]. The IC_50_ value of *Withania somnifera* was 350, 250 and 200 μg/mL when cells were treated for 24 h, 48 h and 72 h, respectively [33]. Our findings revealed that the IC_50_ value of the herbal mixture was higher than for *Helichrysum caespititium*, including some medicinal plants used for contraception in South Africa. Similarly, medicinal plants investigated for contraception in South Africa showed higher acute toxicity than the herbal mixture. For example, an aqueous root bark extract of *Securidaca longipedunculata* orally administered to rats had a lethal dose (LD_50_) of 771 mg/kg [34].

Several South African medicinal plants have been investigated for their contraceptive efficacy using both in vitro and in vivo test systems [7]. For example, the contraceptive efficacy of 300 mg/kg aqueous extracts of *Pouzolzia mixta* was investigated using three groups and eight animals per group [16]. Only two animals did not fall pregnant, a finding that is relatively similar to our result. However, the absence of significant contraceptive effect by the tested herbal mixture does not conclusively translate to its ineffectiveness. Several factors are known to influence the pharmacological response of medicinal plants [20,22,35]. For instance, the type of extracting solvent is known to strongly influence the quality and quantity of phytochemicals [21], which may determine the resultant biological activities [36,37]. In the current study, the choice of water as the extracting solvent was influenced by the need to mimic the approach used in traditional medicine. However, water extracts are known to often exert weak biological activities due to the indiscriminate nature of the extracting phytochemical constituents, which may exert diverse and undesired biological effects [18,37].

## 5. Conclusions

In this study, a preliminary phytochemical profile of the tested herbal mixture used as a contraceptive among indigenous people was generated. Of particular interest was the presence of two phenolic compounds that are known for their contraceptive effects. The safety of the herbal mixture was established using both in vitro and in vivo models. However, the herbal mixture had a limited contraceptive effect at the tested doses. In addition, there was no correlation between applied doses of the herbal mixture extract in relation to the contraception effect and weight of the tested animals. The limited contraceptive efficacy may be related to the use of one solvent (water extract) in the current study. The absence of contraceptive efficacy by the herbal mixture does not mean it is ineffective, as it has a long history of use in the study area. Future studies may entail a range of solvents for the extraction process and a varying range of doses for testing. It will also be pertinent to establish the effect of the herbal mixture on various biochemical parameters involved in contraception.

## Figures and Tables

**Figure 1 plants-11-00193-f001:**
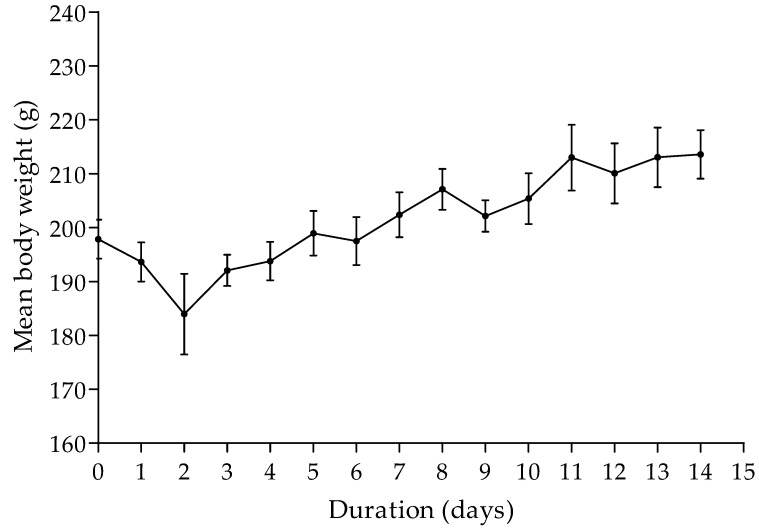
The average weight of animals measured over a period of 14 days. Values are mean (standard error, n = 3).

**Figure 2 plants-11-00193-f002:**
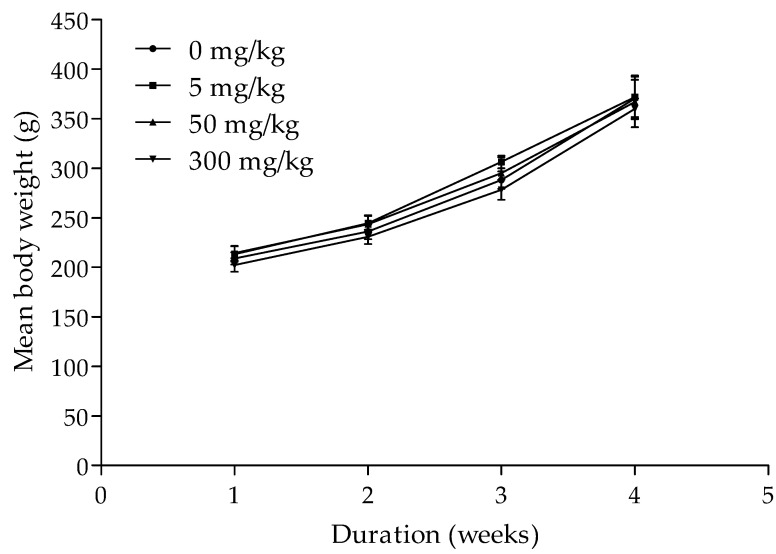
Mean body weight (g) of rats treated with varying doses of aqueous herbal extract weekly over a four-week duration. Values are mean (standard error, n = 7).

**Table 1 plants-11-00193-t001:** Herbal mixture used for contraception by the Batswana traditional health practitioners in Ngaka Modiri Molema District Municipality, South Africa.

Plant (Family)	Voucher Specimen	Local Name	Parts Used	Preparation, Dosage and Administration of Herbal Mixture
*Bulbine frutescens* (L). Willd. (Xanthorrhoeaceae)	Moroole M.A 2	Makgabenyane	Roots	Dried plant parts of *B. frutescens*, *H. caespititium* and *T. trifidum* were cut into small pieces, mixed and crushed into powder. Thereafter, 100 g of the herbal mixture was added to 2 l of water, heated until boiling point and allowed to cool. For women, aliquot of 250 mL daily drank once a day, three days before or after sex.
*Helichrysum caespititium* (DC.) Harv. (Compositae)	Moroole M.A 1	Phate ya Ngaka	Leaves	
*Teucrium trifidum* Retz. (Lamiaceae)	Moroole M.A 4	Setlhokotlhoko	Leaves	

**Table 2 plants-11-00193-t002:** Compounds detected in an aqueous extract of herbal mixture used for contraception by traditional health practitioners in Ngaka Modiri Molema District Municipality, South Africa.

Peak	Compound	Molecular Formula *	Molar Weight (G/Mol) *	Retention Time (S)	Area (%)
1	2,3-Butanedione	C_4_H_6_O_2_	86.09	130.9	0.0904
2	Butanoic acid, methyl ester	C_5_H_10_O_2_	102.131	136.5	0.0184
3	Acetic acid	C_2_H_4_O_2_	60.052	314.7	5.9440
4	Unknown 1	-	-	316.3	2.8014
5	2-Propanone, 1-hydroxy-	C_3_H_6_O_2_	74.078	631.9	0.8804
6	Unknown 2	-	-	669.9	7.4327
7	Unknown 3	-	-	674.4	19.942
8	2,3 Butanediol	C_4_H_10_O_2_	90.122	679.6	44.660
9	Unknown 4	-	-	680.2	1.7356
10	Pyrrrolidin-1-acetic acid	-	-	695.1	0.5558
11	Unknown 5	-	-	849.3	0.8718
12	2,5-Dimethyl-4-hydroxy-3(2H)-Furanone	C_6_H_8_O_3_	128.13	1003.6	1.5595
13	Unknown 6	-	-	1079.4	2.3000
14	Phenol, 4-ethyl-	C_8_H_10_O	122.164	1163.4	0.2300
15	2-Methoxy-4-vinylphenol	C_9_H_10_O_2_	150.17	1219.1	2.6723
16	Glycerin	C_3_H_8_O_3_	92.09	1232.9	4.8424
17	Unknown 7	-	-	1298.1	0,4270
18	Benzaldehyde, 3-methyl-	C_8_H_8_O	120.15	1308.2	1.1407
19	Unknown 8	-	-	1549.8	0.3190
20	Unknown 9	-	-	1584	0.97792
21	Benzeneethanol, 4-hydroxy-	C_8_H_10_O_2_	138.163	1725.2	0.0030054

* Molecular formula and molecular weight were obtained from National Institute of Standards and Technology (NIST) (webbook.nist.gov, accessed 30 November 2021) and National Library of Medicine (NIH) pubchem.nih.gov (accessed 30 November 2021).

## Data Availability

We have included all data related the experiment in the manuscript.

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
