# Peer review of "Phytochemical Profile, Safety and Efficacy of a Herbal Mixture Used for Contraception by Traditional Health Practitioners in Ngaka Modiri Molema District Municipality, South Africa"

_plants, 2022, doi:10.3390/plants11020193_

Round 1
Reviewer 1 Report
Thank you authors for crosschecking the efficacy of traditionally used medicinal plants for their antifertility activities. The paper is well written and the methodology is sound. Results and conclusions are nicely drawn and references are well cited. The paper would serve the global message if the authors make additional discussion by using references of India, China and Himalayan countries. The discussion/paper would be of global significance in doing so. Please find some comments and suggestions in the attached file. I congratulate authors for presenting this high quality paper.
Thank you and ll the best!
Reviewer 2 Report
Dear Authors,
Authors examined about profile of an traditional medicine and effects on animal BW and cell growth.
I recommend to publish after minor revision in below.
1. please insert SD or SE in figure1 and 2.
2. please revise Table 2. There are some compounds repeated. They are mistake. Please write the manuscript carefully.
Best,
Reviewer 3 Report
This Communication aims to investigate the contraceptive potential of an herbal mixture with Bulbine frutescens (roots), Helichrysum caespititium (leaves) and Teucrium trifidum (leaves), used for contraception by traditional health practitioners in South Africa.
The study assessed the phytochemical profile, cytotoxicity, acute oral toxicity and efficacy of the herbal mixture.
The Materials, Methodology of the assays and the Results are well described and very clear for the comprehension and reproducibility of the work, presented with three Figures (one of which is supplementary, two Tables, and the most important findings are well connected and discussed with references to other studies in the Discussion.
The results revealed that the herbal mixture extract proved to be safe but had limited contraceptive efficacy at the tested doses. The results are preliminary, but very important for the validation of traditional medicine.
The recommendation will be to accept the manuscript for publication in the present form.
Reviewer 4 Report
The work is interesting and brings new elements to knowledge. However, it needs to be significantly improved. In particular, I am asking for:
1) adding key information from the authors' previous work (reference 23), especially for providing contraceptive efficacy of pure extracts,
2) in paragraph 2.2 - please give information on equal mass of herbs in prepared mixture,
3) Table 1 (5 column) - last sentence is missible, authors used rats, in text are women,
4) Table 2 - please add for each compound mass to charge ratio (m / z),
5) LD50 is not clear; please explain lethal dose above 2000 mg / kg (lines 249-254),
6) Figure 1 - please add standard deviations,
7) paragraph 3.4 - in my opinion 56-58% variation is insignificant, so please change analysis; please add data of new Table 3 in this paragraph,
8) conclusions - please compare current data to previously obtained; please show directions of new studies in this subject.
Round 2
Reviewer 4 Report
Revised version of manuscript looks well. Authors corrected/explained all my doubts or comments.